# Jatrorrhizine Suppresses Murine-Norovirus-Triggered N-GSDMD-Dependent Pyroptosis in RAW264.7 Macrophages

**DOI:** 10.3390/vaccines11010164

**Published:** 2023-01-12

**Authors:** Ming Fu, Nini Chen, Yanhe Zhou, Sidong Chen, Wanfu Xu, Sitang Gong, Lanlan Geng

**Affiliations:** 1Department of Gastroenterology, Guangzhou Women and Children’s Medical Center, Guangzhou Medical University, Guangzhou 510623, China; 2State Key Laboratory of Virology, Wuhan Institute of Virology, Center for Biosafety Mega-Science, Chinese Academy of Sciences, Wuhan 430071, China

**Keywords:** MNV, jatrorrhizine, NLRP3, inflammasome, pyroptosis

## Abstract

Human norovirus (HNV) is one of the emerging and rapidly spreading groups of pathogens and the main cause of epidemic viral gastroenteritis globally. Due to a lack of in vitro culture systems and suitable animal models for HNV infection, murine norovirus (MNV) has become a common model. A recent study showed that MNV activates NLRP3 inflammasome leading to pyroptosis. Jatrorrhizine (JAT) is a natural isoquinoline alkaloid isolated from *Coptis Chinensis*, which has been proven to have antibacterial, anti-inflammatory, and antitumor effects. However, whether JAT has an effect on norovirus gastroenteritis and the underlying molecular mechanism remain unclear. Here, we found that JAT could ameliorate NLRP3-N-GSDMD-dependent pyroptosis induced by MNV infection through inhibiting the MAPKs/NF-κB signaling pathways and decrease MNV replication in RAW264.7 macrophages, suggesting that JAT has the potential to be a therapeutic agent for treating norovirus gastroenteritis.

## 1. Introduction

Noroviruses (NoVs) are non-enveloped, linear, single-stranded, positive-sense RNA viruses belonging to the genus Norovirus of the family Caliciviridae [1]. Noroviruses are classified into seven genogroups (GI–GVII) based on the sequences of RdRp and the VP1 gene, of which GI, GII, and GIV can infect humans and cause illness [2]. HNV is a leading cause of sporadic acute diarrhea episodes and outbreaks of acute gastroenteritis across all age groups [3,4,5]. These viruses are highly infectious, as even a few particles can cause disease, and infected individuals shed high loads of the virus [6]. The infection is primarily spread via the fecal–oral route with transmission occurring through person-to-person transmission, the ingestion of contaminated food or water, or through contact with contaminated media [7,8]. Due to the high infectivity and efficient transmission, newly emerged strains of norovirus can cause global epidemics. HNV symptoms are mild and self-limited in most cases, while severe and long-term symptoms can occur in the elderly and immunocompromised [9,10], suggesting that a comprehensive understanding of the pathogenesis of HNV is necessary. HNV cannot readily be grown in a cell culture and no small animal model of human norovirus infection is available, both of which represent a major obstacle to HNV research [5]. Because of its shared characteristics with HNV strains and cultivable in mice as well as in murine dendritic cells and macrophages, MNV is used as a common model to further our knowledge of HNV [11,12].

NLRP3 inflammasome, assembled by NLRP3, ASC, and caspase1, plays a crucial role in host defense against pathogen invasion and homeostasis [13]. However, overactivation of the NLRP3 inflammasome contributes to the development of various inflammatory diseases, including Alzheimer’s disease, cryopyrin-associated periodic syndromes (CAPS), gout, autoinflammatory diseases, and atherosclerosis [14,15,16]. A two-step process of priming (signal 1) and assembly (signal 2) is required for NLRP3 inflammasome activation [17,18]. Pattern recognition receptor signaling triggers signal 1, subsequently leading to the transcriptional activation of NLRP3, caspase1, ASC, pro-IL-1β, and pro-IL-18 through nuclear factor-κB (NF-κB)-dependent pathways. Pathogen-associated molecular patterns (PAMPs) and damage-associated molecular patterns (DAMPs) can induce the activation signal (signal 2). Numerous molecular or cellular events, such as the efflux of potassium ions (K^+^) or chloride ions (Cl^−^), the flux of calcium ions (Ca^2+^), lysosomal disruption, mitochondrial dysfunction, metabolic changes, and trans-Golgi disassembly, are involved in the activation of NLRP3 inflammasome assembly, such as mitochondrial dysfunction and reactive oxygen species (ROS) generation and lysosomal damage [19,20]. Due to the activation of the NLRP3 inflammasome, it induces self-cleavage and activation of pro-caspasse1, leading to the maturation and release of the proinflammatory cytokines IL-1β and IL-18. In addition, activated caspase-1 also cleaves gasdermin D (GSDMD) and releases its N-terminal domain (N-GSDMD), which migrates to the cell membrane and forms pores, resulting in the release of all kinds of cellular contents, including IL-1β and IL-18, then activating a strong inflammatory response, and inducing the inflammatory cell death known as pyroptosis [13,21]. Therefore, NLRP3 inflammasome activation must be strictly controlled, and its comprehensive mechanism needs to be fully studied. A recent study showed that MNV infection induced NLRP3 inflammasome activation and N-GSDMD-driven pyroptosis [22].

As an isoquinoline alkaloid, jatrorrhizine (JAT) is a bioactive metabolite found in common medicinal plants, including *Berberis vernae* Schneid., *Tinospora sagittata* (Oliv.) Gagnep., and *Coptis Chinensis* Franch [23,24,25,26], which have been used for the treatment of gastroenteritis, abdominal pain, or diarrhea [27,28,29,30]. One of the effective ways to obtain plant metabolites is chemical synthesis. At present, the synthesis of jatrorrhizine has been achieved [31]. A few studies have shown that JAT has anti-inflammation activity, and a mechanistic analysis demonstrated that JAT suppressed TNF-α-stimulated activation of NF-κB and MAPKs leading to the downregulation of proinflammatory cytokines IL-1β and IL-18 [32,33,34]. However, the role of JAT in regulating the viability and function of MNV-infected macrophages remains unclear.

On the basis of these findings, the effect of JAT on the survival of the MNV-infected RAW264.7 macrophages was investigated. Due to ASC deficiency in RAW264.7 macrophages [35], we constructed a stable ASC-expressing RAW264.7 cell line and used it in this study. The current study showed that JAT at 0 to 160 μM had no significant cytotoxic effect on RAW264.7 cells. In addition, JAT treatment significantly reduced the cytotoxic damage of RAW264.7 cells caused by MNV infection. We also found that JAT could ameliorate NLRP3-N-GSDMD-dependent pyroptosis caused by MNV infection in RAW264.7 macrophages through inhibiting MAPKs/NF-κB signaling pathways and inhibiting MNV replication, suggesting that JAT has the potential to be a therapeutic agent for treating norovirus gastroenteritis.

## 2. Materials and Methods

### 2.1. Cell Culture and Virus

The RAW264.7 murine macrophages were purchased from the American Type Culture Collection (ATCC), and the cells used in this study were a stable ASC-expressing line, produced by delivering a lentiviral-vector-inserted complementary DNA of mouse ASC into RAW264.7 macrophages. Briefly, the open-reading frame of ASC was inserted into a pLenti6.3-MCS-IRES2-EGFP plasmid (Miaoling, China) to obtain a pLenti6.3-ASC plasmid. Then, the pLenti6.3-ASC plasmids were transfected into RAW264.7 cells, and ASC^+^ isolates were screened out. The cells were grown in Dulbecco’s modified Eagle medium (DMEM, ThermoFisher, Scoresby, Australia)) with 10% fetal bovine serum (FBS, Thermo Scientific, Sydney, Australia) and 100 U/mL penicillin/streptomycin (Genom, Hangzhou, China) at 37 °C with 5% CO_2_. 

RAW 264.7 macrophages were cultured in 10 cm^2^ sterile cell culture dishes. When 70–90% confluence was reached, 2.5 mL of the culture medium was left and 1.5 mL MNV-1 suspension was added to each dish. Followed by 2 h for viral infection at 37 °C with 5% CO_2_, 8 mL DMEM with 2% FBS was added. After 2–4 days of cultivation, the RAW 264.7 cells were frozen and thawed 3–5 times to release virus particles. After 3 min of centrifugation at 10,000 rpm, the obtained supernatant was then filtered through a 0.45 µm filter (Merck millipore), aliquoted, and stored at −80 °C for later use.

MNV titration was determined by a plaque assay on RAW264.7 macrophages monolayers. RAW 264.7 macrophages were inoculated into 12-well plates at a density of 1 × 10^6^ cells per well in 1 mL of medium. On day 2, monolayer cells were infected with a 10-fold diluted virus series of 100 μL at 37 °C for 2 h. After the inoculum was removed, the cells were covered with a 1 mL medium containing 2% FBS and 1% agarose and cultured for 2–4 days. Then, the cells were fixed with 4% paraformaldehyde for 0.5 h. The agarose was removed, and 0.5% crystal violet solution was added to stain the cells. Plates with 5 to 50 plaques were used to determine the virus titer in PFU. 

### 2.2. MNV Infection and JAT Treatment

RAW264.7 macrophages were seeded in 96-well plates at a density of 4000 cells per well in 100 μL of medium. To explore the effect of JAT (Yuan Ye, B21476, Shanghai, China) on the cell viability, cells were treated with JAT (0, 5, 10, 20, 40, 80, and 160 μM) for 48 h at 37 °C. To explore the effect of JAT on the viability of RAW264.7 macrophages infected with MNV (500 pfu), cells infected with MNV for 24 h were treated with JAT (10, 20, and 40 μM) for 48 h at 37 °C. RAW264.7 macrophages were seeded in 6-well plates at a density of 2 × 10^6^ cells per well in 2 mL of medium to investigate NLRP3 inflammasome activation and pyroptosis caused by MNV infection with or without treatment of JAT.

### 2.3. Cell Viability Assay

Cell viability was detected using a CCK-8 counting kit (Zeta life, K009, San Francisco, USA, CA, USA) according to the manufacturer’s instructions. RAW264.7 cells were plated into a 96-well plate at a density of 4000 cells per well in 100 μL of medium. After treating the RAW264.7 cells or the MNV-infected RAW264.7 cells with JAT for 48 h, 10 μL of CCK-8 solution was added into each per well and incubated for 1.5 h. The absorbance was determined at 450 nm using a microplate reader (Molecular Devices, San Jose, CA, USA). The effect on cell viability was assessed as the percent cell viability compared with the untreated control group, which was assigned 100% viability.

### 2.4. Western Blotting Analysis

The experiments were performed as described previously [36]. RAW264.7 macrophages infected with MNV for 48 h or infected with MNV for 24 h and treated with JAT for 48 h were lysed using a lysis buffer (20 mM Tris (pH7.5), 150 mM NaCl, 1% Triton X-100, sodium pyrophosphate, β-glycerophosphate, EDTA, Na3VO4, and leupeptin) supplemented with a protease inhibitor cocktail (Roche, 11697498001, Mannheim, Germany). Protein concentration was determined by a BCA assay according to the manufacturer’s instructions (Takara Bio, T9300A, Beijing, China), and then the samples were boiled for 10 min with a loading buffer (50 mM Tris-HCl, pH 6.8, 2% SDS, 25% glycerol, 1% DTT). Prepared cell lysates were subjected to 12% SDS-PAGE and transferred to 0.45 µm polyvinylidene difluoride (PVDF) membranes (Millipore, BS-00-2529, Cork, Ireland). The PVDF membrane was blocked with 5% non-fat milk and subsequently incubated overnight at 4 °C with the following primary antibodies: an anti-NLRP3 antibody, an anti-caspase1 antibody, an anti-p-p65 antibody, an anti-ASC antibody, an anti-N-GSDMD, and an anti-β-tubulin antibody. After three washes with TBS-Tween (200 mM NaCl, 50 mM Tris-HCl, 0.1% Tween-20), the membrane was incubated with secondary antibodies (HRP-conjugated goat anti-rabbit IgG (Proteintech, B900210, China) or goat anti-mouse IgG (Proteintech, SA00001-1, China)) for 1 h at room temperature. The antibodies were purchased from Proteintech, Wuhan, China. Protein bands were visualized following incubation with enhanced chemiluminescence (ECL) (Millipore, Billerica, MA, USA).

### 2.5. Enzyme-Linked Immunosorbent Assay (ELISA) of IL-1β and IL-18 in Cell Culture Supernatants

The cell culture supernatants of RAW264.7 macrophages infected with MNV for 48 h or infected with MNV for 24 h and treated with JAT for 48 h were collected and stored at −80 °C for later use. The mouse IL-1β ELISA kit (MEIMIAN, MM-0040M1, Jiangsu, China) and the mouse IL-18 ELISA kit (MEIMIAN, MM-0169M1, Jiangsu, China) were used to analyze the levels of IL-1β and IL-18 in the cell culture supernatants following the manufacturer’s instructions. Briefly, 50 μL of gradient diluted standard or sample was accurately added to each well and incubated for 30 min at 37 °C, and all standards and samples were added in duplicate to the Microelisa Stripplate. After washing five times with 1 × washing buffer, 100 μL of HRP-conjugate reagent was added to each well and incubated 30 min at 37 °C. Washing the plate five times again, chromogen A 50 μL and chromogen B 50 μL were added, and the plate was incubated for 10 min at 37 °C. Finally, 50 μL of stop solution was added to each well to stop the reaction. The signal was quantified at a 450 nm wavelength using a microplate reader (Molecular Devices, Silicon Valley, CA, USA).

### 2.6. Viral RNA Isolation and qRT-PCR Analysis

According to the manufacturer’s instructions, Trizol (Invitrogen, 15596-026, Waltham, MA, USA) was used to extract viral RNA from the cell supernatant of RAW264.7 macrophages. Then, the HiScript II Q RT SuperMix for qPCR (+gDNA wiper) (Vazyme Biotech, R223-01, Nanjing, China) was used to obtain the cDNA. The newly synthesized cDNA was used as the template to amplify the targeted gene. ChamQ SYBR qPCR Master Mix (High ROX Premixed) (Vazyme Biotech, Q341-02, Nanjing, China) was used to conduct a RT-PCR, and the experiment was performed on a CFX Real-Time PCR system (Bio-Rad, California, USA) according to the following conditions: 95 °C for 1 min, followed by 40 cycles of 95 °C for 15 s, 60 °C for 15 s, and 72 °C for 45 s. The following are the primers designed to target the vp1 gene of MNV (5′-CAGGCAGAAACACTCCTAAT-3′ and 5′-GGTAGAAGTACTGCACCCATTCC-3′). The expression difference was calculated on the basis of 2^−ΔCt^ values.

### 2.7. Statistical Analysis

The results are presented as the mean ± SD for at least three independent experiments. Student’s t-tests were used to analyze the data to determine the difference between the two groups. Statistical analysis was performed using GraphPad Prism software (version 8.0, San Diego, CA, USA) and *p* < 0.05 was considered to indicate statistical significance.

## 3. Results

### 3.1. MNV Induces Activation of NLRP3 Inflammasome and Pyroptosis

A recent report showed that MNV could activate NLRP3 inflammasome leading to N-GSDMD-driven pyroptosis in a mouse model [22]. As expected, MNV infection increased the protein levels of NLRP3, pro-caspase1, ASC, cleaved-caspase1 and N-GSDMD (Figure 1A), as well as secreted IL-1β and IL-18 (Figure 1B,C), in RAW264.7 macrophages, suggesting that MNV infection activated NLRP3 inflammasome. Moreover, the CCK-8 assay showed that MNV infection increased cell death (Figure 1D), which is consistent with the increased protein level of N-GSDMD.

### 3.2. JAT Decreases Cytotoxicity Caused by MNV

JAT is a natural isoquinoline alkaloid isolated from *Coptis Chinensis*, which has been proven to have an anti-inflammatory effect. To ascertain whether JAT could alleviate cytotoxicity caused by MNV infection, MNV-infected RAW264.7 macrophages were treated with or without JAT for 48 h. The results showed that JAT did not affect RAW264.7 macrophages’ viability and proliferation at concentrations of 5, 10, 20, 40, 80, and 160 μM (Figure 2A), and JAT could significantly inhibit the reduction in RAW264.7 macrophage viability caused by MNV infection in a dose-dependent manner (Figure 2B).

### 3.3. JAT Suppresses GSDMD Cleavage through Inhibiting NLRP3 Inflammasome Activation

We then explored the mechanism by which JAT suppressed cytotoxicity caused by MNV infection. The results showed that N-GSDMD was significantly downregulated in the JAT treatment group, as well as the protein levels of NLRP3, cleaved-caspase1, and secreted IL-1β and IL-18 (Figure 3A–C), indicating that JAT inhibited the activation of NLRP3 inflammasome and thus reduced the cleavage of GSDMD to N-GSDMD.

### 3.4. JAT Inhibits MNV-Induced Activation of MAPKs/NF-κB Signaling

Mechanistic studies have shown that JAT could suppress the activation of NF-κB and MAPKs stimulated by TNF-α, leading to the downregulation of proinflammatory cytokines [32,33], and MAPKs and NF-κB play important roles in the activation of the NLRP3 inflammasome [37,38]. To investigate whether JAT suppressed NLRP3 inflammasome activation by inhibiting MAPKs/NF-κB signaling, we analyzed the related protein levels, and the results showed that MNV infection resulted in significant activation of MAPK/NF-κB signaling as indicated by the increased phosphorylation levels of ERK, p38, JNK, and p65 in a dose-dependent manner, which was inhibited by JAT treatment except JNK (Figure 4).

### 3.5. JAT Inhibits the Replication of MNV

As some alkaloid structures have strong antiviral effects, we intended to investigate whether JAT has an impact on the replication of MNV. The cell culture supernatants of RAW264.7 macrophages infected with MNV for 24 h and treated with JAT for 48 h were collected, and the RNA level of MNV capsid protein VP1 was tested. The results showed that JAT significantly downregulated the VP1 RNA level in a dose-dependent manner at 10, 20, and 40 μM, indicating that JAT could efficiently inhibit MNV replication (Figure 5A). To figure out whether the independent antiviral activity of JAT leads to the inhibition of inflammatory responses, we further collected the cell culture supernatants and cell lysates of RAW264.7 macrophages infected with MNV for 24 h and treated with JAT for 24 h. The results showed that the VP1 RNA level had not decreased (Figure 5B), while the phosphorylation levels of ERK, p38, and p65 were still inhibited by JAT (Figure 5C), indicating that the anti-inflammatory activity of JAT played a major role at 24 h of treatment, and the antiviral activity also played an important role at 48 h. 

## 4. Discussion

Lacking in vitro culture systems and suitable animal models is the major obstacle to HNV research. Because MNV shares characteristics with HNV strains and can be cultured in mice, mouse dendritic cells, and macrophages, MNV is used as a generic model to study the genetic and biochemical characteristics of NoVs [39]. Dubois et al. provided new insights into the mechanisms of norovirus-induced inflammation and cell death. They revealed that the activation of the NLRP3 inflammasome and subsequent N-GSDMD-driven pyroptosis were contributors to immunopathology in susceptible STAT1-deficient mice induced by MNV infection [22]. Our results verified that MNV infection increased cell death and the N-GSDMD protein level of RAW264.7 macrophages and activated the NLRP3 inflammasome.

JAT is a bioactive metabolite in common medicinal plants, which has been proven to have an anti-inflammatory effect and is used for the treatment of gastroenteritis, abdominal pain, or diarrhea [27,28,29]. However, the role of JAT in regulating MNV-infected macrophages’ viability is unclear. Here, JAT was shown to decrease the cytotoxicity of MNV infection in RAW264.7 macrophages, and the increased protein level of N-GSDMD decreased after treatment with JAT, indicating that JAT could efficiently inhibit MNV-infection-induced N-GSDMD-driven pyroptosis. Furthermore, we found that JAT could decrease the protein levels of NLRP3, cleaved-caspase1, and secreted IL-1β and IL18 in a dose-dependent manner, which were increased by MNV infection, indicating that JAT suppressed MNV-infection-induced pyroptosis through inhibiting NLRP3 inflammasome activation. Previous studies have shown that NF-κB and MAPKs activate the NLRP3 inflammasome [37,38] while JAT could suppress TNF-α-stimulated activation of NF-κB and MAPKs [32,33,34]. We then investigated and found that MNV infection increased the phosphorylation levels of ERK, p38, JNK, and p65, which were inhibited by JAT treatment except JNK, indicating that JAT inhibited NLRP3 inflammasome activation through suppressing the activation of NF-κB and MAPKs (p38 and ERK). As some alkaloid structures have strong antiviral effects [40,41,42], we explored the anti-MNV effect of JAT and found that JAT could efficiently inhibit MNV replication. 

## 5. Conclusions

In conclusion, this work has contributed to a potential treatment for MNV-induced acute gastroenteritis. We identified that JAT treatment inhibited NLRP3 inflammasome activation through suppressing the activation of NF-κB and MAPKs (p38 and ERK) in MNV-infected RAW264.7 macrophages, and then decreased the cleavage of GSDMD to N-GSDMD and ultimately ameliorated N-GSDMD-driven pyroptosis. Moreover, JAT could efficiently inhibit MNV replication. Taken together, the current study demonstrates that JAT exerts a protective effect against MNV infection in RAW264.7 macrophages by inhibiting NF-κB/MAPKs/NLRP3/N-GSDMD-driven pyroptosis and inhibiting MNV replication, suggesting a potential role of JAT in treating MNV-induced acute gastroenteritis.

## Figures and Tables

**Figure 1 vaccines-11-00164-f001:**
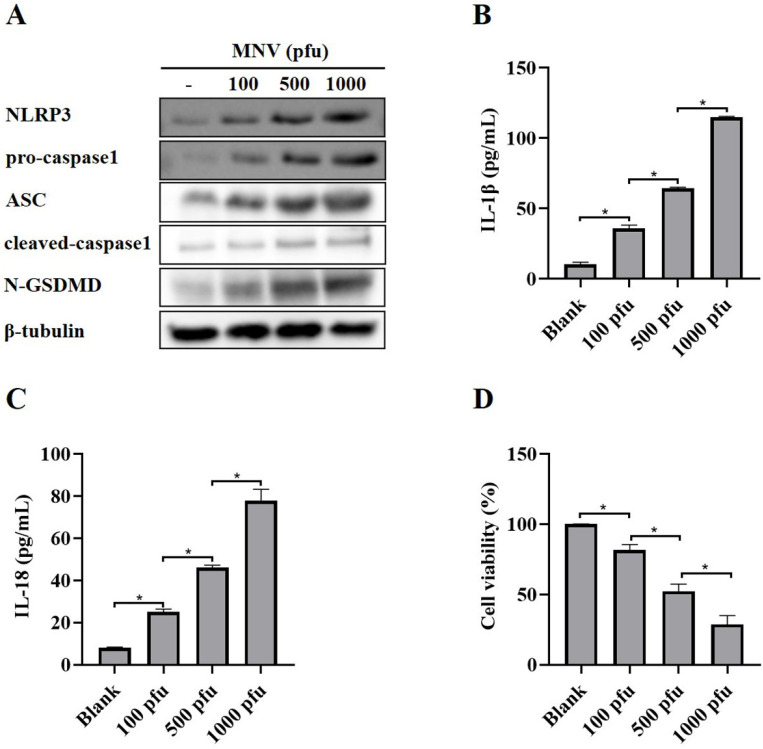
NLRP3 activation and pyroptosis in RAW264.7 macrophages induced by MNV infection. RAW264.7 macrophages infected with MNV for 48 h were lysed, and the culture supernatants were collected. (**A**) Cell lysates were immunoblotted for NLRP3, pro-caspasse1, ASC, cleaved-caspase1, N-GSDMD, and β-tubulin. (**B**,**C**) The cell culture supernatants were analyzed for secreted IL-1β and IL-18 levels by ELISA. (**D**) The CCK-8 assay detected the cell viability of RAW264.7 macrophages infected by MNV. * *p* < 0.05.

**Figure 2 vaccines-11-00164-f002:**
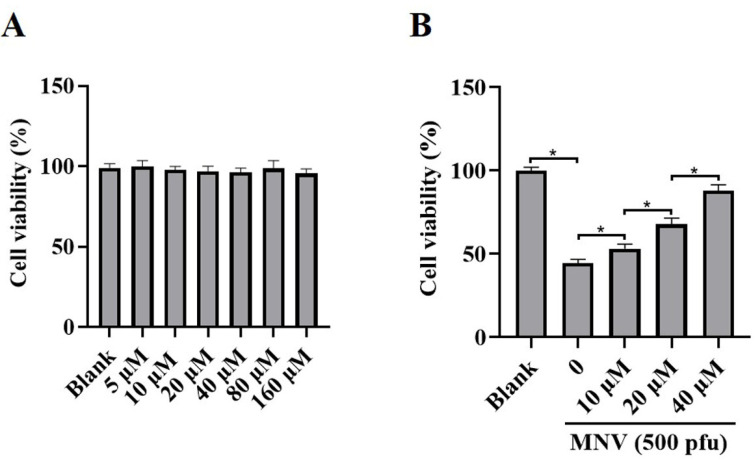
JAT reduces the cytotoxicity of MNV on RAW264.7 macrophages. RAW264.7 macrophages were infected with MNV for 24 h and treated with JAT for 48 h. (**A**) The cell viability of the RAW264.7 macrophages treated with different concentrations of JAT (5, 10, 20, 40, 80, and 160 μM) was detected by CCK-8 assay. (**B**) The cell viability of the MNV-infected RAW264.7 macrophages in the absence or presence of JAT (10, 20 and 40 μM) was detected by a CCK-8 assay. * *p* < 0.05.

**Figure 3 vaccines-11-00164-f003:**
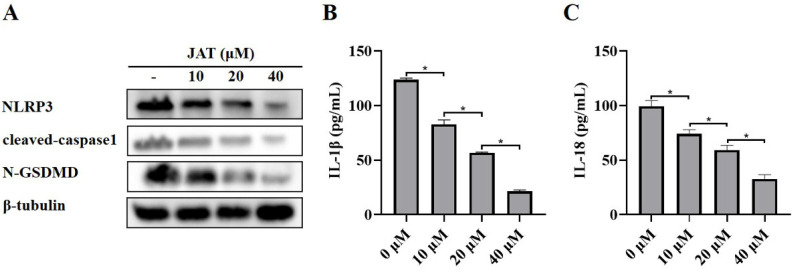
JAT inhibits MNV-infection-induced NLRP3 inflammasome activation in RAW264.7 macrophages. RAW264.7 macrophages infected with MNV for 24 h and treated with JAT for 48 h were lysed, and the culture supernatants were collected. (**A**) The cell lysates were immunoblotted for NLRP3, cleaved-caspase1, N-GSDMD, and β-tubulin. (**B**,**C**) The cell culture supernatants were analyzed for secreted IL-1β and IL-18 levels by ELISA. * *p* < 0.05.

**Figure 4 vaccines-11-00164-f004:**
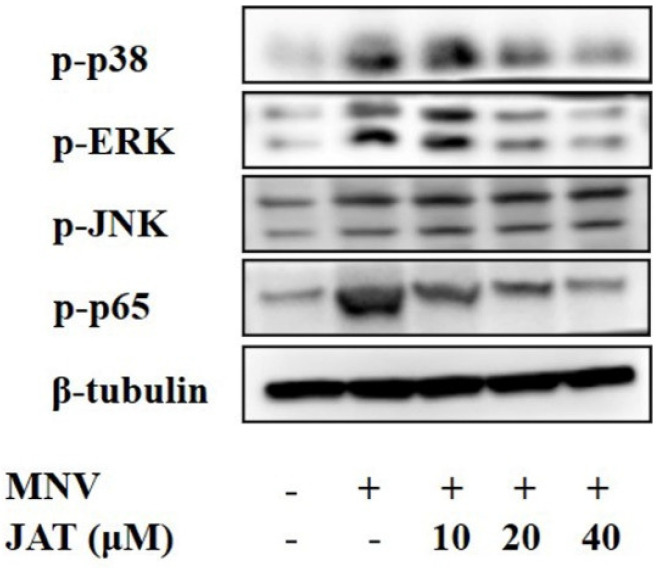
JAT inhibits MNV-infection-induced activation of MAPKs/NF-κB signaling. RAW264.7 macrophages infected with MNV for 24 h and treated with JAT for 48 h were lysed. Cell lysates were immunoblotted for phosphorylated MAPKs (p38, ERK and JNK), p65, and β-tubulin.

**Figure 5 vaccines-11-00164-f005:**
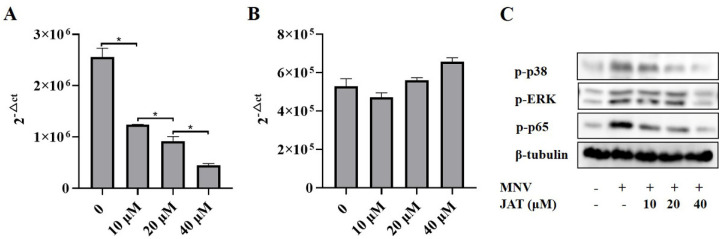
JAT inhibits MNV replication. The cell culture supernatants of RAW264.7 macrophages infected with MNV for 24 h and treated with JAT for 48 h were collected. (**A**) MNV RNA was extracted from the collected cell supernatant and the VP1 RNA level was tested by qRT-PCR. The cell culture supernatants and cell lysates of the RAW264.7 macrophages infected with MNV for 24 h and treated with JAT for 24 h were collected. (**B**) MNV RNA was extracted from the collected cell supernatant and the VP1 RNA level was tested by qRT-PCR. (**C**) Cell lysates were immunoblotted for phosphorylated MAPKs (p38 and ERK), p65 and β-tubulin. * *p* < 0.05.

## Data Availability

The primary data used to support the findings of this study are available from the corresponding author upon request.

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
