# Peer review of "Jatrorrhizine Suppresses Murine-Norovirus-Triggered N-GSDMD-Dependent Pyroptosis in RAW264.7 Macrophages"

_vaccines, 2023, doi:10.3390/vaccines11010164_

Round 1

Reviewer 1 Report

In this interesting manuscript, the researchers described their research on jatrorrhizine, murine norovirus, and macrophage-like RAW264.7 cells. Specifically, they investigated the ability of jatrorrhizine to suppress the replication of murine norovirus and to inhibit virus-induced NLRP3 inflammasome activation and pyroptosis in the macrophages.

The reviewer respectfully offers the following comments for consideration by the authors. The reviewer hopes these comments will be helpful.

1. The scientific literature indicates that RAW264.7 cells lack ASC, which is needed for the activation of the NLRP3 inflammasome, yet the researchers demonstrated the presence of ASC in Figure 1A. Did the researchers use a different cell line (other than RAW264.7), or did they use RAW264.7 cells that had been genetically manipulated so they would express ASC? The researchers should address these points in their manuscript.

Below is a link to an article indicating that RAW264.7 cells lack ASC:

https://journals.aai.org/jimmunol/article/180/11/7147/84652/P2X7-Receptor-Differentially-Couples-to-Distinct

2. More information is needed in the Materials and Methods section so the reader can understand how the studies were done and others will be able to repeat the experiments. A few examples are given below.

2a. The researchers should describe how they prepared their stocks of murine norovirus and how the plaque assay was performed in macrophage-like RAW264.7 cells (Materials and methods, section 2.1).

2b. The number of cells and volume of the medium used in the 96-well plates and in the 6-well plates should be stated for each type of experiment. In addition, more information is needed about how the researchers infected the RAW264.7 cells with murine norovirus, and how soon after infection the jatrorrhizine was added (Materials and methods, sections 2.2 and 2.3).

2c. For Western blotting (section 2.4), the authors should state how soon after murine norovirus infection the RAW264.7 cells were lysed.

2d. The timing of the different types of experiments performed should be noted in the Materials and Methods section. It would also be helpful if this information was briefly stated in each figure legend.

2e. The source of the jatrorrhizine should be given. Was it purchased, obtained from other researchers, or prepared and purified by the authors?

3. The legend for Figure 3 should be modified to reflect the data shown in the figure. Figure 3 shows data on NLRP3 inflammasome activation and release of IL-1β and IL-18; this figure does not show data on macrophage viability.

4. In a number of places in the manuscript, the language should be modified to make the meaning clearer. The reviewer notes only a few examples below; however, there are others.

4a. In line 105, don’t the authors mean “enzyme-linked immunosorbent assay” rather than “enzyme-linked immune-absorption assay”?

4b. In line 148, don’t the authors mean “macrophage viability” rather than “macrophage activity”?

4c. Lines 191-193, lines 199-200, and lines 219-220 in the manuscript are not clear.

Author Response

In this interesting manuscript, the researchers described their research on jatrorrhizine, murine norovirus, and macrophage-like RAW264.7 cells. Specifically, they investigated the ability of jatrorrhizine to suppress the replication of murine norovirus and to inhibit virus-induced NLRP3 inflammasome activation and pyroptosis in the macrophages.

The reviewer respectfully offers the following comments for consideration by the authors. The reviewer hopes these comments will be helpful.

  1. The scientific literature indicates that RAW264.7 cells lack ASC, which is needed for the activation of the NLRP3 inflammasome, yet the researchers demonstrated the presence of ASC in Figure 1A. Did the researchers use a different cell line (other than RAW264.7), or did they use RAW264.7 cells that had been genetically manipulated so they would express ASC? The researchers should address these points in their manuscript.

Below is a link to an article indicating that RAW264.7 cells lack ASC:

https://journals.aai.org/jimmunol/article/180/11/7147/84652/P2X7-Receptor-Differentially-Couples-to-Distinct

RE: Sorry to overlooked it. The RAW264.7 cells used here is a stable ASC-expressing line, and we have revised it (Materials and methods, section 2.1).

  1. More information is needed in the Materials and Methods section so the reader can understand how the studies were done and others will be able to repeat the experiments. A few examples are given below.

2a. The researchers should describe how they prepared their stocks of murine norovirus and how the plaque assay was performed in macrophage-like RAW264.7 cells (Materials and methods, section 2.1).

RE: We have revised it.

2b. The number of cells and volume of the medium used in the 96-well plates and in the 6-well plates should be stated for each type of experiment. In addition, more information is needed about how the researchers infected the RAW264.7 cells with murine norovirus, and how soon after infection the jatrorrhizine was added (Materials and methods, sections 2.2 and 2.3).

RE: We have revised it.

2c. For Western blotting (section 2.4), the authors should state how soon after murine norovirus infection the RAW264.7 cells were lysed.

RE: We have revised it.

2d. The timing of the different types of experiments performed should be noted in the Materials and Methods section. It would also be helpful if this information was briefly stated in each figure legend.

RE: Thanks for your suggestion, and we have revised it.

2e. The source of the jatrorrhizine should be given. Was it purchased, obtained from other researchers, or prepared and purified by the authors?

RE: The jatrorrhizine was purchased, and we have revised it.

  1. The legend for Figure 3 should be modified to reflect the data shown in the figure. Figure 3 shows data on NLRP3 inflammasome activation and release of IL-1β and IL-18; this figure does not show data on macrophage viability.

RE: We have revised it.

  1. In a number of places in the manuscript, the language should be modified to make the meaning clearer. The reviewer notes only a few examples below; however, there are others.

4a. In line 105, don’t the authors mean “enzyme-linked immunosorbent assay” rather than “enzyme-linked immune-absorption assay”?

RE: We have revised it.

4b. In line 148, don’t the authors mean “macrophage viability” rather than “macrophage activity”?

RE: We have revised it.

4c. Lines 191-193, lines 199-200, and lines 219-220 in the manuscript are not clear.

RE: We have revised it.

Reviewer 2 Report

Fu et al., explore the antiviral activity of jatrorrhizine against MNV infection and find that it possesses a modest antiviral activity in a RAW macrophage-like cell line. The authors then show that drug treatment inhibits MNV-triggered inflammasome and NFKB pathways. The authors claim that their mechanism of action of the drug is that it directly inhibits MNV-triggered NFKB activation which in-turn is required for inflammasome activation. The authors provide no direct evidence supporting this hypothesis. Of course, an independent antiviral activity of the compound which lowers viral titers would also be expected to result in inhibition of inflammatory responses. The authors must provide additional data to prove their mechanism of action. Further, more details on methods are needed in the figure legends. For example, in figure 1 what is the MOI used to infect cells and at what time point post infection are these data collected?

Author Response

Fu et al., explore the antiviral activity of jatrorrhizine against MNV infection and find that it possesses a modest antiviral activity in a RAW macrophage-like cell line. The authors then show that drug treatment inhibits MNV-triggered inflammasome and NF-κB pathways. The authors claim that their mechanism of action of the drug is that it directly inhibits MNV-triggered NF-κB activation which in-turn is required for inflammasome activation. The authors provide no direct evidence supporting this hypothesis. Of course, an independent antiviral activity of the compound which lowers viral titers would also be expected to result in inhibition of inflammatory responses. The authors must provide additional data to prove their mechanism of action. Further, more details on methods are needed in the figure legends. For example, in figure 1 what is the MOI used to infect cells and at what time point post infection are these data collected?

RE: Thanks for the good advice. Based on the current data, it is hard to figure out which function of JAT contributes to the inhibition of inflammatory responses. Furtherly, the cell culture supernatants and cell lysates of RAW264.7 macrophages infected with MNV for 24 h and treated with JAT for 24 h were collected, and we found that the VP1 RNA level was not decreased, while the phosphorylation levels of ERK, p38, and p65 were still inhibited by JAT, indicating that the anti-inflammatory activity of JAT played a major role. Of course, antiviral activity also contributes to the inhibition of inflammatory responses when JAT was treated for 48 hours.  

Round 2

Reviewer 1 Report

The authors have revised their manuscript and addressed most of the comments in my first review.  Below are my comments on their revised manuscript.

1. In the materials and methods section, the researchers have now stated that they used a genetically modified RAW264.7 cell line.  Additional specific information about this cell line should be included in the methods section or added in an appendix. 

2.  In the materials and methods section, the authors have added information about the cell number per well; however the volume of medium used was not given.

3.  Some parts of the manuscript would still benefit from editing to make the English language and meaning clearer.

Author Response

The authors have revised their manuscript and addressed most of the comments in my first review.  Below are my comments on their revised manuscript.

1. In the materials and methods section, the researchers have now stated that they used a genetically modified RAW264.7 cell line.  Additional specific information about this cell line should be included in the methods section or added in an appendix. 

RE: Additional information about the RAW264.7-ASC cells is supplemented in the materials and methods section.

2.  In the materials and methods section, the authors have added information about the cell number per well; however the volume of medium used was not given.

RE: We have revised it (Lines 90, 98, 103 and 108).

3.  Some parts of the manuscript would still benefit from editing to make the English language and meaning clearer.

RE: Thanks for your suggestion, and we have revised it.

Reviewer 2 Report

The authors have added additional data demonstrating that at 48 HPI of infection and treatment that intracellular levels of MNV RNA are unchanged while extracellular levels are decreased. This does not suggest that anti-inflammatory and antiviral activities of Jatrorrhizine are separable but instead provides a nice piece of mechanistic evidence that their compound blocks the virus lifecycle downstream of viral genome replication such as assembly or egress. The authors should modify their manuscript to make it clear that there isn’t sufficient evidence to sperate these two phenotypes. The additional detail the authors added to the methods section is sufficient.

Author Response

The authors have added additional data demonstrating that at 48 HPI of infection and treatment that intracellular levels of MNV RNA are unchanged while extracellular levels are decreased. This does not suggest that anti-inflammatory and antiviral activities of Jatrorrhizine are separable but instead provides a nice piece of mechanistic evidence that their compound blocks the virus lifecycle downstream of viral genome replication such as assembly or egress. The authors should modify their manuscript to make it clear that there isn’t sufficient evidence to sperate these two phenotypes. The additional detail the authors added to the methods section is sufficient.

RE: Thanks for your suggestion. The anti-inflammatory and antiviral activities of Jatrorrhizine both contributed, making it difficult to sperate these two phenotypes. We have revised the manuscript to make it clear (Lines 222-223).
